# {COMPANYNAME}11K: An Unsupervised Representation Learning Dataset for Arrhythmia Subtype Discovery

## Abstract

We release the largest public electrocardiogram (ECG) dataset of continuous raw signals for representation learning containing 11 thousand patients and 2 billion labelled beats. Our goal is to enable semi-supervised ECG models to be made as well as to discover unknown subtypes of arrhythmia and anomalous ECG signal events. To this end, we propose an unsupervised representation learning task, evaluated in a semi-supervised fashion. We provide a set of baselines for different feature extractors that can be built upon. Additionally, we perform qualitative evaluations on results from PCA embeddings, where we identify some clustering of known subtypes indicating the potential for representation learning in arrhythmia sub-type discovery.

## 1 Introduction

Arrhythmia detection is presently performed by cardiologists or technologists familiar with ECG readings. Recently, supervised machine learning has been successfully applied to perform detection of certain types of arrhythmia (Hannun et al., 2019; Yıldırım et al., 2018; Mincholé & Rodriguez, 2019; Porumb et al., 2020).

However, there may be ECG anomalies that warrant further investigation because they do not fit the morphology of presently known arrhythmia. We seek to use a data driven approach to finding these differences that cardiologists have anecdotally observed, which motivates the representation learning potential of this data.

Our data is collected by the {DEVICENAME}™, a single-lead heart monitor device from {COMPANYNAME}(Paquet et al., 2019). The raw signals were recorded with a 16-bit resolution and sampled at 250Hz with the {DEVICENAME}™in a modified lead 1 position. The wealth of data this provides us can allow us to improve on the techniques currently used by the medical industry to process days worth of ECG data, and perhaps to catch anomalous events earlier than currently possible. All data is made public[1].

The ethics institutional review boards at the {UNIVERSITY} approved the study and release of data #{STUDYID}

### 1.1 Objective

We want to improve the state-of-the-art of automated arrhythmia detection via representation learning. Ideally, this representation should preserve as much information about the underlying true heart function as possible. Such representations and learned feature extractors can improve downstream tasks which require more complicated features than what is typically extracted to predict major cardiac issues. More concretely, we are proposing a semi-supervised challenge on ECG data.

While an objective method to evaluate such a representation would be to measure its performance on tasks of interest, the way to perform best on such an evaluation would be to directly run a supervised learning task on those objectives. However, in certain circumstances, like training a neural

---

[1]Data available: URL

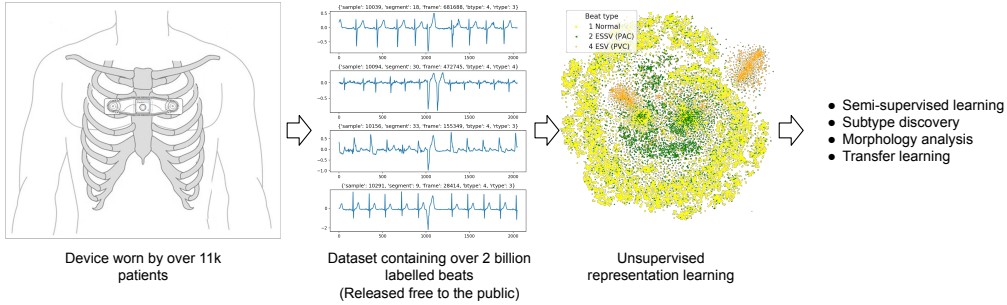

Figure 1: Overview of the project.

network, for example, doing so results in a loss of information about the input (Tishby & Zaslavsky, 2015). The process may remove information vital to the discovery of new sub-types. We will see an example of this in §5.2.

Extracting features which can predict outcomes of not just arrhythmia is also an existing field of study (Lerma & Glass, 2016; Karpagachelvi et al., 2010), and can benefit from learned feature extractors based on this data.

## 2 RELATED WORK

ECG (or sometimes known as EKG) signals are collected by electrocardiograph machines. These machines traditionally have 10 electrodes, resulting in 12-lead ECG data. These can be thought of as a 12 channel signal that provides additional data about the heartbeat, but allows for only short periods of data capture due to the cumbersome nature of these machines, and are not sufficient for capturing rarer events that happen over time.

One of the first open dataset of ECG signals was the MIT-BIH dataset, created in 1979 (Moody & Mark, 2001). They "expected that the availability of a common database would foster rapid and quantifiable improvements in the technology of automated arrhythmia analysis." The MIT-BIH is still in use today with just 47 subjects. However, Shah & Rubin (2007); Guglin & Thatai (2006) found that computer predictions during that time were fraught with errors.

Later, data collection efforts improved leading to the creation of many small specific datasets (Goldberger et al., 2000). The MIMIC-III Waveform Database (Johnson et al., 2016) contains 67,830 waveform records from 30,000 ICU patients. These samples are at a higher sampling rate and with more leads. However, they are only recorded for short periods of time. The ECG-ViEW II dataset (Kim et al., 2017) aims to be a freely available dataset of ECG records together with clinical data for 461,178 patients. Instead of raw signals, only beat information is included: RR interval, PR interval, QRS duration, etc.[2] Figure 2 shows the basic ECG form where a letter identifies each aspect of the beat: P, Q, R, S, and T. The STAFF III Database (Pablo Martínez et al., 2017) contains 104 patients under an acutely induced myocardial ischemia. This includes pre, during, and post catheter insertion.

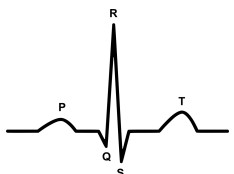

Figure 2: QRS regions for an ECG heartbeat signal.

More recently, single-lead wearable devices provided much larger amounts of data than before. As these devices could be worn for throughout the day, over a period of a couple of weeks, machine learning had much more data to work with. Rajpurkar et al. (2017) created an annotated training dataset of ECG signals consisting of 30,000 patients (Turakhia et al., 2013). The authors' approach, and the follow up work claim that their automated models perform at the level of trained cardiologists (Hannun et al., 2019). However, their data has not been made publicly available.

---

[2]The letters indicate the interval between the two events. For example, PR interval is the time in between the P and R event, while an RR interval is the length of time between two Rs in a different heartbeat.

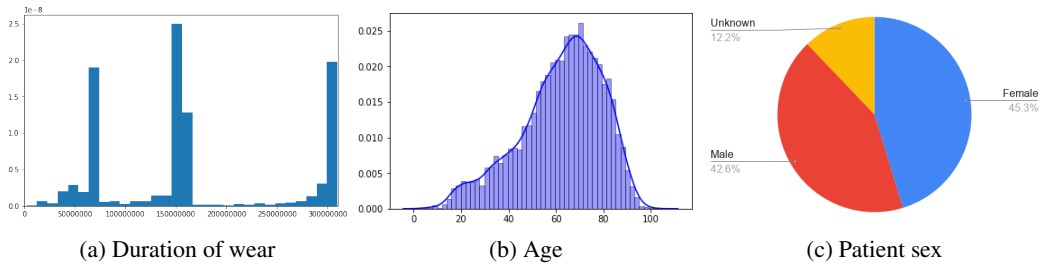

(a) Duration of wear       (b) Age       (c) Patient sex

Figure 3: Demographics of the patients in the data.

## 3 PRIVACY CONCERNS

Our data has been made completely anonymous so that no reidentification is possible. For the sake of precision, no name or serial number of any kind is associated with a particular data set.

### 3.1 HEARTBEATS AS BIOMETRICS

There are attempts to use ECG signal data as a biometrics to identify someone. This brings risk of re-identification from our anonymized signal data.

A paper by Salloum & Kuo (2017) claims high performance but the evaluation does not seem very controlled. A company called Nymi also aims to use a wearable ECG as a method to authenticate users.

When viewed in the context of other literature, the claim that ECG is a reliable method of authentication seems to be diminished. For example, Song et al. (2017) explores alternative ways to sense cardiac motion (movement of the heart, which they say is the identifying aspect), stating that ECG "biosignals are not related to cardiac motion, in which case indirect or incomplete cardiac characterization will compromise the advantages of cardiac motion as a biometric".

Israel & Irvine (2012) state "Unlike fingerprint and face, the heartbeat data could contain health-related information as well as the personal identification information. This suggests a need for greater care in the collection, storage, and transmission of such data." Additionally, they say that ECG has several limitations that must be overcome before they can be used as a biometric. Specifically, that (1) it requires a sufficient number of samples to identify an individual because the signal does not contain much information, (2) the combination of varying environments and individual yields a unique signal, (3) A target's emotional state also requires intra-individual normalisation, and (4) a change in the contact location can reduce the ability to identify someone.

## 4 {COMPANYNAME}11K DATASET

The dataset is processed from data provided by 11,000 patients who used the {DEVICENAME}[TM]device predominantly in Ontario, Canada, from various medical centers. While the device captures ECG data for up to two weeks, the majority of the prescribed duration of wear was one week. Figure 3a shows the distribution over duration of wear in the unprocessed data.

It should be noted that since the people who wear the device are patients, the dataset does not represent a true random sample of the global population. For one, the average age of the patient is $62.2\pm17.4$ years of age. Furthermore, whereas the {DEVICENAME}[TM]can be worn by any patient, it is mostly used for third line exam[3], so the majority of records in the dataset exhibit arrhythmias. No particular effort has been done on patient selection except data collection has been conducted over years 2017 and 2018. Figure 3c shows the distribution over age and gender.

The data is analysed by {COMPANYNAME}'s team of 20 technologists who performed annotation on proprietary analysis tools. When the data is first extracted from the device, beat detection is performed automatically. A first technologist looks at the record as soon as possible to quickly send

---

[3]Most patients were prescribed {DEVICENAME}[TM]by a tertiary referral hospital or care centre

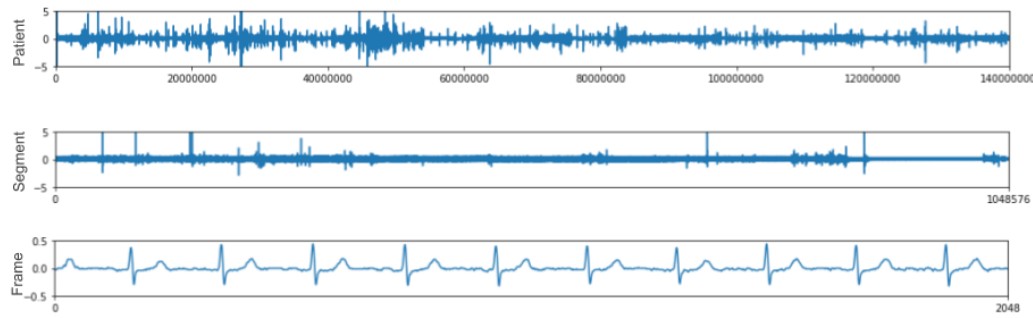

Figure 4: ECG data at different levels of the hierarchy. From top to bottom, a full patient record, a segment, and a frame.

a feedback on the severity of the case. A second technologist then analyses the record labelling beats and rhythms (these will be further elaborated in Section 5.1) performing a full disclosure analysis i.e. he / she sees the whole recording. The types of labels are described in more detail in Section 5.1. Finally, the analysis is approved by a senior technologist.

To prepare the data, we segment each patient record into segments of $2^{20} + 1$ signal samples ( $\approx 70$ minutes). This longer time context was informed by discussions with technologists: the context is useful for rhythm detection. We made it a power of two with a middle sample to allow for easier convolution stack parameterisation. From this, we randomly select 50 of the segments and their respective labels from the list of segments. The goal here is to reduce the size of the dataset while maintaining a fair representation of each patient. In the training data we remove the labels for 80% of the patients. For the remaining 20%, half will be kept for the semi-supervised task, while another half will remain as test data for evaluation. Further details of nomenclature and statistics of the unprocessed and processed data can be found in Table 1.

We describe in further detail the different levels of hierarchy we have separated the data into:

**Patient level (3-14 days)** At this level, the data can capture features which vary in a systematic way and not isolated events, like the placement of the probes or patient specific noise.

**Segment level (approximately 1 hour)** A cardiologist can look at a specific segment and identify patterns which indicate a disease while ignoring noise from the signal such as a unique signal amplitude. Looking at trends in the segment help to correctly identify arrhythmia as half an hour provides the necessary context to observe the stress of a specific activity.

**Frame level (approximately 8 seconds)** At this level, the data can capture features about the beat as well as the rhythm.

While we have provided baselines only for frame-level features in this paper, we believe that processing the data with these levels of hierarchy results in some grouping information that could be leveraged to attain better results.

## 5 UNSUPERVISED REPRESENTATION LEARNING TASK

While the processed data includes labelled beat and arrhythmia information, we propose an *unsupervised* representation learning challenge to the community.

The goal of this data is to develop unsupervised representations of the ECG signal which can aid in two aspects:

1. Improve the performance of supervised tasks by using the learned representations.
2. Identify unknown subtypes of disease by studying the clustering of the representations.

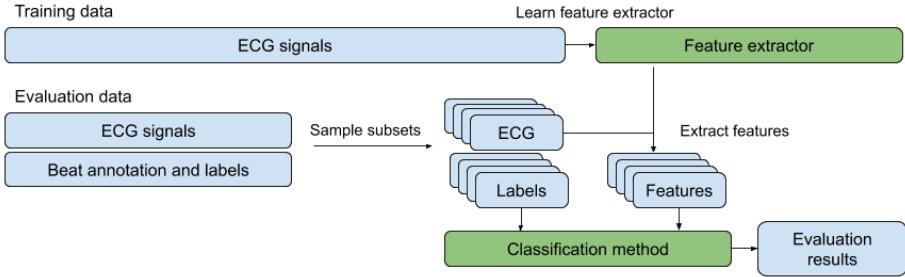

Figure 5: Diagram detailing the training and evaluation pipeline for the representation learning task. We have provide different methods in this paper for the blocks colored in green.

These issues are addressed in quantitative and qualitative evaluations in the next two sections. The focus of this section studies the frame level embeddings which are typically enough for cardiologists to interpret.

### 5.1 QUANTITATIVE EVALUATION

For the quantitative evaluation we will benchmark common unsupervised algorithms in a semi-supervised setting to establish base quality. We make all code and models public in order to facilitate reproducibility and future work[4].

The evaluation consists of predicting the beat and rhythm for each frame in a hold out set (samples id's $> 10,000$). The beat task is to predict if a frame contains all normal beats or contains at least one premature ventricular contractions (PVC) or premature atrial contraction (PAC) anywhere in a frame. Classifying a beat alone regardless of its surrounding beats can be challenging as, for example, a PAC is an abnormal beat only because it appears too soon and disrupts the rhythm (frequency). Furthermore, a PAC beat has the same shape as a normal beat, so taken alone, you can nearly not make the difference with a normal beat. The model will need to construct features about the nearby beats as well.

The second task is to predict the rhythm type given a frame. For a given frame the classification method must predict if the rhythm is normal, atrial fibrillation (AFib)[5], or atrial flutter, based on the input representation. AFib is indicated by irregular RR intervals, no distinct P waves and usually variable intervals between two atrial activations (Vollmer et al., 2018). Flutter appears as a saw-tooth pattern of R waves. Recognising both patterns require contexts larger than a single beat. These labels

---

[4]https://github.com/shawntan/icentia-ecg

[5]AFib is a controversal rhythm as cardiologists do not agree on the minimum duration. 8 second frames might not be sufficient to make such a decision.

| Term | Definition |
|------|-----------|
| Segment | Fixed length contiguous region of a signal. |
| Sample | As used in signal processing: A scalar value representing the amplitude of the signal in time. |
| Event | A specific arrhythmia occurring. |

(a) Glossary of terms

| Statistic | # (units) |
|-----------|-----------|
| Number of Patients | 11,000 |
| Number of labeled beats | 2,774,054,987 |
| Sample Rate | 250Hz |
| Frame size | $2^{11} + 1 = 2,049$ samples |
| Segment size | $2^{20} + 1 = 1,048,577$ samples |
| Total number of frames | 1,084,314 |
| Total number of segments | 542,157 |
| Dataset Size | 271.27GB |

(b) Dataset Statistics

Table 1: Reference tables

| Beat labels | Count | | Rhythm Labels | Count |
|---|---|---|---|---|
| Normal | 174,249 | | NSR (Normal Sinusal Rhythm) | 261,377 |
| Premature Atrial Contractions | 58,780 | | AFib (Atrial Fibrillation) | 13,056 |
| Premature Ventricular contractions | 44,835 | | AFlutter (Atrial Flutter) | 3,330 |
| (a) Beat labels in the test set | | | (b) Rhythm labels in the test set | |

Table 2: Label counts in the test subset (patients 9000-10999). Each frame has a label. Only 2 types of labels are provided. Only these meaningful labels are used for evaluation and presented to the classifier.

are annotated at the beat level. If a beat is a beat-level anomaly, this will be labelled at the beat where the event occured. If a beat is within an anomalous rhythm period, the beats within the rhythm would all be labelled with the corresponding rhythm type.

Both these tasks used in a supervised classification problem as a proxy for evaluating the usefulness of extracted features for detecting such events.

Figure 5 shows the pipeline for our evaluation method. Code to perform this evaluation in a consistent fashion is made available online for replicating the results and implementing new methods. The bulk of the training data does not come with beat annotation and labels, and can be used to train or fit a feature extraction method. The evaluation consists of sampling $N$ frames from the test set and computing representations using the *feature extractor*. 50% of the data is then used to train a *classification method* and then evaluated on the held out 50%. Two classification models are used: (1) A k-nearest neighbors (KNN) method with $k = 3$, and (2)an MLP method, which consists of 4 layers of dimensions 1024, 1024, 512, and 512. The MLP model was trained for 10 epochs with Adam optimizer. We applied dropout (Srivastava et al., 2014) to prevent overfitting.

Each representation is learned without knowledge of the tasks — the feature extraction model is not updated during training of the classifier. We provide the evaluation results for the following baseline feature extraction methods:

**Principal Components Analysis (PCA)**  computes the principal components from 30k examples from the training data. Then projects the test data onto 100, 50, or 10 principal components.

**Fast Fourier Transform (FFT)**  computes a Fourier transform representing the magnitude of frequencies between 1Hz and 125Hz (Cooley & Tukey, 1965).

**Periodogram**  computes an estimate of the power spectral density using Welch's method (Welch, 1967).

**BioSPPy**  identifies each beat using the detection algorithm by Carreiras et al. (2015) and computes the mean and standard deviation then concatenates them together to form the representation.

**Autoencoder**  (Hinton, 1990) comprises of 2 MLPs, an encoder with an input size of 2049, a hidden layer of dimension 200, and a bottleneck representation of 100 dimensions. The decoder has the same architecture in reverse. There are residual connections before each non-linearity, and a batch normalization (Ioffe & Szegedy, 2015) is performed at the bottleneck layer. The model is trained for 3 epochs with Adam (Kingma & Ba, 2014) at a learning rate $10^{-4}$ with the L2 loss.

Our hope is that evaluation using a semi-supervised setting on known arrythmia labels (e.g. premature atrial contraction, premature ventricular contraction) and the various rhythm labels (e.g. atrial fibrillation, atrial flutter) is a sufficient proxy for the quality of a representation — that these representations will prove useful for discovering unknown disease subtypes. Two models are used to evaluate the representations. We utilize small numbers of samples ($N = 1000$ and $N = 20000$) for evaluation to simulate the situation where a small cohort of patients is augmented using the unlabelled data we provide. Balanced accuracy is used to compute performance because there is a large imbalance between classes. If a model is to predict the same class for all samples the maximum balanced accuracy will be 0.33. We expect that this also becomes a source of noise at $N = 1000$

| Model | KNN | | | | MLP | | | |
| --- | --- | --- | --- | --- | --- | --- | --- | --- |
| | N = 1000 | | N = 20000 | | N = 1000 | | N = 20000 | |
| | Beat | Rhythm | Beat | Rhythm | Beat | Rhythm | Beat | Rhythm |
| Random | 0.33±0.02 | 0.33±0.00 | 0.33±0.01 | 0.33±0.01 | 0.33±0.01 | 0.33±0.01 | 0.33±0.02 | 0.33±0.00 |
| Raw Sequence | 0.44±0.03 | 0.33±0.00 | 0.61±0.01 | 0.34±0.00 | 0.54±0.03 | 0.38±0.08 | **0.67±0.01** | 0.33±0.01 |
| PCA $R^{100}$ | 0.50±0.04 | 0.33±0.01 | **0.65±0.01** | 0.34±0.01 | 0.55±0.04 | 0.36±0.07 | **0.67±0.01** | 0.33±0.00 |
| PCA $R^{50}$ | **0.51±0.03** | 0.33±0.00 | 0.64±0.01 | 0.34±0.00 | 0.55±0.04 | 0.34±0.06 | 0.64±0.01 | 0.33±0.00 |
| PCA $R^{10}$ | 0.46±0.02 | 0.34±0.01 | 0.52±0.01 | 0.34±0.00 | 0.47±0.03 | 0.40±0.06 | 0.50±0.01 | 0.33±0.00 |
| FFT | 0.48±0.02 | **0.37±0.03** | 0.53±0.01 | 0.36±0.01 | 0.50±0.04 | 0.41±0.09 | 0.54±0.01 | 0.33±0.00 |
| Periodogram | 0.43±0.02 | 0.35±0.03 | 0.47±0.01 | 0.36±0.01 | 0.49±0.03 | **0.44±0.10** | 0.53±0.01 | 0.33±0.00 |
| BioSPPy mean beat | 0.35±0.02 | 0.34±0.01 | 0.38±0.01 | **0.39±0.01** | 0.40±0.03 | 0.34±0.08 | 0.40±0.01 | 0.33±0.00 |
| AE (Random init) | 0.41±0.03 | 0.33±0.00 | 0.53±0.01 | 0.34±0.00 | 0.45±0.02 | 0.36±0.08 | 0.55±0.00 | 0.33±0.00 |
| AE | 0.51±0.04 | 0.34±0.01 | 0.64±0.01 | 0.34±0.01 | **0.56±0.02** | 0.38±0.07 | 0.66±0.01 | 0.33±0.00 |

Table 3: Performance on a semi-supervised task computed as balanced accuracy. Given a random subset of labels from the training set predict the labels in the test set. Evaluated over 10 random subsets.

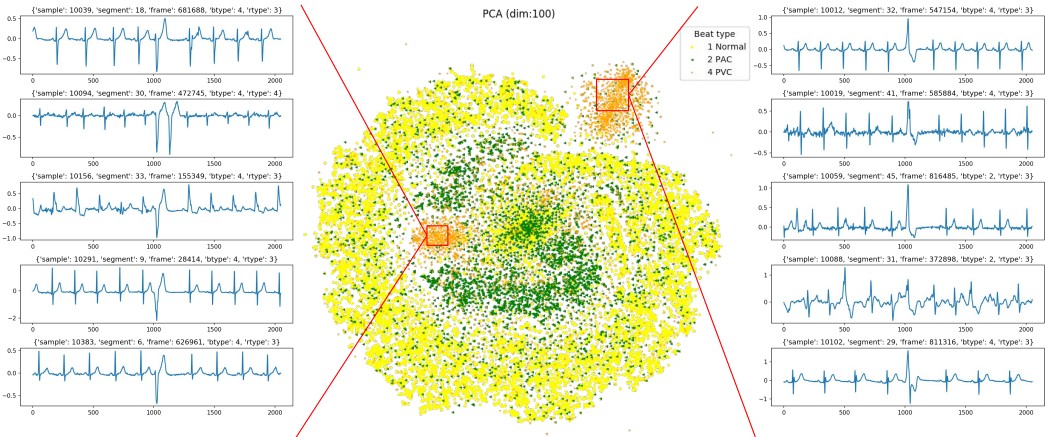

Figure 6: An analysis of the specific clusters resulting from the PCA features of 100 dimensions visualized with a t-SNE. 40,000 example frames were randomly sampled from the test data.

because an underrepresented class has a large impact in the performance if random predictions get a few samples right by chance.

The results are shown in Table 3. Currently autoencoders are not able to perform as well as we expected. PCA is able to perform the best at beat detection when using the KNN model while the MLP is able to predict better using the raw signal. One surprise is that rhythm detection is difficult. It is possible that, because the Periodogram and FFT captures periodicity in the signal, it performs better than the other feature extraction methods. Work by Vollmer et al. (2018) has shown that it is possible in a supervised setting.

The results also shows the issues with using MLPs as a classification method for this task. MLPs typically requires more data points for training, and this issue shows up in the $N = 1000$ case, where there is a higher variance in the accuracy for each subset. The effect is even larger in rhythm classification, where the classes are imbalanced, resulting in huge variations in the *balanced accuracy*. When more data is available ($N = 20000$), variance is lower. As the ultimate purpose of this task is to learn better representation of the ECG signal, having a powerful parametric models like an MLP that works well only on higher instance counts may be offloading the representation learning to the classification method, which, as we alluded to before, is not favourable in our setting.

## 5.2 QUALITATIVE EVALUATION

Medical literature has discussed multiple types of PVC (Kanei et al., 2008; Phibbs, 2006). PVCs can be monomorphic or multimorphic (have different morphologies). Additionally, PVCs can also be multifocal and manifest in a different shape. In a multi-lead setting, when arising from the right ventricle, it has a dominant S wave in one particular lead but has a dominant R wave if generated from the left ventricle Phibbs (2006).

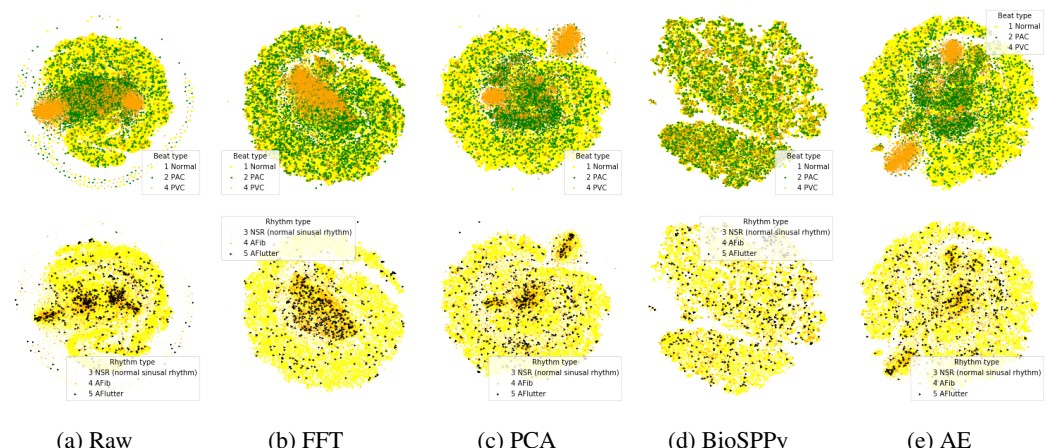

|   (a) Raw   |   (b) FFT   |   (c) PCA   |   (d) BioSPPy   |   (e) AE   |

Figure 7: t-SNE plot of embeddings produced by different frame-level encoders. Colors represent the three basic labels. Each plot is computed using the same 20,000 frame examples encoded using each method and then having a t-SNE applied.

We investigate the clustering of the signals by looking at the PCA encoding of 40,000 frames using a t-SNE in Figure 6. The plots clearly show two clusters of PVC that we can interpret as two different morphologies of this arrhythmia. We note that these are easy to see because of the different colors we use to highlight the points, but there seems to be remaining clusters that have not been analysed. The correlation between having two clusters for PVCs and PVCs being multimorphic aspect may be of interest to medical researchers to further explore clusters in this space created by different feature extractors.

Many other encoding methods, shown in Figure 7, also show clustering related to PVC and PAC. Notably FFT and BioSppy do not break the PVCs into two clusters. Although we can observe rhythm having some grouping it does not appear significant in the quantitative evaluation.

Such analysis is similar to what is done by Kachuee et al. (2018). However, in that work the features were constructed using a supervised task.

## 6 CONCLUSION

Single-lead heart monitors like the {DEVICENAME}$^{\text{TM}}$are increasingly common, and have the potential for cardiologists to learn much more about arrhythmia and related heart diseases. However, this amount of data means manual analysis is no longer practical.

Machine learning has been widely deployed in the medical field by training a model to predict the right diagnosis based on human expert labels. Supervised learning serves well as an assistant in medical field; however, it hardly provides information beyond human knowledge. Additionally, certain human body signals can be very complex and imply non-linear features that cannot be easily identifiable manually. At present, representation learning methods have a potential in disentangling complex features, and potentially, unveil new signal structures of certain diseases which can correlate with clinical presentations.

By releasing this dataset, we believe that we can leverage unsupervised representation learning expertise to not only help to enable training models with lower number of samples, but potentially find new diseases and identify patterns associated with them.

We have proposed an evaluation pipeline for learning a feature extractor and evaluating extracted features using known arrhythmia as a proxy to measure the usefulness of the features. In addition, we have provided baseline results for *frame*-level representations under different feature extraction methods. Our data preparation makes a three level hierarchy available — the *segment* and *patient* level grouping of data. While we did not provide baselines that exploit this, future work that can

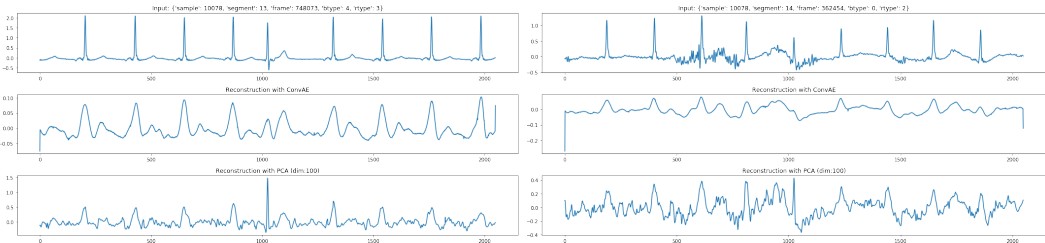

Figure 8: Reconstructions using the AE and PCA[100]. Two samples are shown, one for each column. The input is shown on the top followed by the AE and then PCA.

take advantage of this context to extract better representations, and perhaps, find more interesting structure in the representation space. We also believe that this dataset can serve as a benchmark in other areas of machine learning, such as anomaly and outlier detection and hierarchical sequence modelling.

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
