# OpenReview forum: "{COMPANYNAME}11K: An Unsupervised Representation Learning Dataset for Arrhythmia Subtype Discovery"
_ICLR.cc/2020/Conference — Reject_

### Official Review · AnonReviewer2 · 2019-10-21
**Official Blind Review #2**

**Rating:** 3

**Review:**

Overall the paper is well written and organized and is an application of various paradigms to extract most relevant features for Arrhythmia Subtype classification.
Plan to release the largest (claimed) public ECG data set of continuous raw signals for representation learning containing 11 thousand patients and 2 billion labelled beats.
Our goal is to enable semi-supervised ECG models to be made as well as to discover unknown sub types of arrhythmia and anomalous ECG signal events.

The stated intended goal though is the discovery of new  Arrhythmia Sub types (title), “automated arrhythmia detection” (1.1 Objective).
A definition/example of a regular ECG signal, one presenting arrhythmia, various types of arrhythmia signal forms, would be welcome for the paper to fit this audience and be self-contained.

This work suffers from many drawbacks:
-	A definition/example of a regular ECG signal, one presenting arrhythmia, various types of arrhythmia signal forms, would be welcome for the paper to fit this audience and be self-contained.

-	Given the intended audience in ICLR, the application domains specificities are not well explained. Many concepts are not introduced clearly:
      o	ECG models, sub types of arrhythmia, anomalous ECG signal events.
      o	In related work section:
            	Paragraph 2: more “leads”, RR interval, PR interval, QRS duration, induced myocardial ischemia.
            	Paragraph 3: single-lead wearable devices.
      o	In 3 privacy concerns: heartbeats as biometrics section/ Paragraph 2:
            	alternative ways to sense “cardiac motion”.
      o	In 4 {companyname} 11k data set section:
            	Paragraph 2: “third line exam”.
            	Paragraph 3: “beats and rhythms”, these are fundamental to understanding graphical results provided.
      o	In 5.1 quantitative evaluation section:
            	Paragraph 3: “ .. irregular RR intervals, no distinct P waves and usually variable intervals between two atrial activations.
            	Page 6:  “ .. high-level abnormality labels”
      o	5.2 qualitative evaluation section
            	Paragraph 1:  “ .. has a dominant S wave in V1 lead.”
      o	In 6 conclusion section:
            	“ Single-lead heart monitors” .

-	Figures are barely decipherable
      o	In Figure 6:
            	What are ESSV and ESV, btype and rtype. Could you give a list of known subtypes?
            	Is this for PVC data only?

-	Many paragraphs are not clear to us:
      o	Introduction/Paragraph 2:
            	“While cardiologists are able to see these differences, it is hard to conclude that they are ‘real’ by finding the same anomalous signal across multiple time points and patients without a data driven approach.”
      o	In 3 privacy concerns: heartbeats as biometrics section/ Paragraph 3:
            	Contrary to 1), 3) & 4), is (2) “.. the expression of environmental variables on the heartbeat data is unique to the individual”, a limitation to ECG to being considered a biometric measure?
      o	In 4 {companyname} 11k dataset section:
            	Paragraph 4:  “we segment each patient record into segments of 220 +1 signal samples ( ~70 minutes). Care to explain the rationale?
            	Last paragraph: “ .. we believe that processing the data with these levels of hierarchy results in some grouping information that could be leveraged to attain better results.”. This is a multi-scale approach. But do you have any medical (application domain) knowledge that would justify/hint to using such approach?
      o	Figure 5:
            	Only Beats are labelled. No Rhythm labelling?
      o	Table 2:
            	“Only 2 types of labels are provided”. Are these beats and Rhythms? What are anomalies considered in the study then?
      o	In 5.1 quantitative evaluation section:
            	Paragraph 2: “..PAC is an abnormal beat only because it appears too soon and disrupts the rhythm (frequency). Furthermore, a PAC beat has the same shape as a normal beat, so taken alone, you can nearly not make the difference with a normal beat.”. A graphical representation would be welcome.
            	Paragraph 3: “..Both require a representation that will compose a representation showing the difference between beats over time.”.
      o	qualitative evaluation section
            	Paragraph 2: “We note that these are easy to see because of the different colors we use to highlight the points, but there seems to be remaining clusters that have not been analysed.”. The clusters are not evident to us even after purposefully colored.


**Experience Assessment:**

I have read many papers in this area.

**Review Assessment: Checking Correctness Of Derivations And Theory:**

I assessed the sensibility of the derivations and theory.

**Review Assessment: Checking Correctness Of Experiments:**

I carefully checked the experiments.

**Review Assessment: Thoroughness In Paper Reading:**

I read the paper thoroughly.

---

> ### Author Response · Authors · 2019-11-13
> **We greatly appreciate the detailed remarks to improve readability.**
>
> It seems like the major reason for the reject decision was that our paper was not accessible to the ICLR audience. We went over the paper and with your comments addressed the areas that were too much on the cardiology side. Please let us know if we have not addressed all your issues and changed your decision on this paper.
>
> > Given the intended audience in ICLR, the application domains specificities are not well explained. Many concepts are not introduced clearly
>
> We added further clarifications on some of the terminology, but we avoided inundating the paper with too many details. We added a brief description what “leads” are in the ECG context, and explained what intervals were.
>
> > Figures are barely decipherable
>
> We have corrected the figures with ones of a higher resolution. We've also removed the additional acronyms that may have made things confusing.
>
>
> > Paragraph 2: “We note that these are easy to see because of the different colors we use to highlight the points, but there seems to be remaining clusters that have not been analysed.”. The clusters are not evident to us even after purposefully colored.
>
> The clusters have been pointed out by red lines and samples in them have been visualised as signals.

---

### Official Review · AnonReviewer3 · 2019-10-21
**Official Blind Review #3**

**Rating:** 3

**Review:**

This paper describes a large-scale ECG dataset that the authors intend to publish. Along with the to-be-released dataset, the authors also provide some unsupervised analysis and visualization of the dataset.

The data is collected using the author’s company’s single lead ECG devices. The dataset is collected from 11,000 patients and contains over 2.7 billion peaks. According to the cited works in section 2, the scale of this proposed dataset is unprecedented and will be beneficial to the ECG community.

I think this is an interesting dataset for the ECG machine learning community. Some recent advances in this field are based on non-public dataset. However, this dataset seems to require additional curation before it is ready.



Questions.
In section 3, the authors discuss the potential concerns regarding privacy, what are the measures taken by the authors when collecting & distributing this dataset? I didn’t find this question addressed in section 3.

Like the authors stated in 2nd paragraph under section 4, this dataset is biased in the sense that it is collected from patients. The subjects are in the age group of 62.2 +/- 17.4. Do the authors recognized various chronicle conditions that often appear in that age group? Is that information part of the to-be-released dataset?

Since this work focuses on releasing  a dataset, I find the current experiment section unnecessary & somewhat unrelated. What would be interesting to see is whether the authors use ML techniques to facilitate the curation of the dataset. For example, in paragraph 3 under section 4, it says each segment data is labeled by 3 technologists. How many technologist are involved in total? and what measure are taken to address the variances among different technologists?


**Experience Assessment:**

I have read many papers in this area.

**Review Assessment: Checking Correctness Of Derivations And Theory:**

N/A

**Review Assessment: Checking Correctness Of Experiments:**

N/A

**Review Assessment: Thoroughness In Paper Reading:**

I read the paper at least twice and used my best judgement in assessing the paper.

---

> ### Author Response · Authors · 2019-11-13
> **Thank you for your comments. Baseline experiments are crucial for dataset papers.**
>
> It seems that you appreciate this work but decided to reject this paper due to missing details and the inclusion of an experimental section. Baseline experiments for dataset papers provide a useful benchmark for the community to compare with and guide further work on the topic. Discussions with cardiologists have guided us to the set of evaluation metrics that we’ve described.
> Existing dataset papers do this (e.g. https://openreview.net/forum?id=r1lYRjC9F7, https://openreview.net/forum?id=r1l73iRqKm).
>
> We have also addressed the issues with missing details below and we hope that you will reconsider your decision.
>
> > this dataset seems to require additional curation before it is ready.
> Could the reviewer clarify what additional curation would make the dataset ready?
>
> > what are the measures taken by the authors when collecting & distributing this dataset?
>
> We’ve anonymised the data to comply with the ethics review board requirements by our university (this was mentioned in the paper). We have added additional details about the steps we’ve taken to the paper.
>
> > Do the authors recognized various chronicle conditions that often appear in that age group?
>
>
> That information will not be included in the released dataset. With regard to chronic conditions, do you mean we should be including the types of chronic conditions for the general population of where the patients come from? We are unable to include conditions associated with each patient.
>
> > How many technologists are involved in total? and what measure are taken to address the variances among different technologists?
>
> In total there are 20 technologists (we’ve updated the paper to include this). For each record, three technologists were involved. For more details see Section 4 paragraph 3.
> We note that the annotations created by these technologists were then used in a medical diagnosis of the patient involved.

---

### Public Comment · ~Ilya_Kotlov1 · 2020-10-02
**dataset availability**

Hi. Could you maybe upload the data also in another source, partitioning it to some amount of parts  ?
I'm trying to download it as a torrent and it is almost impossible due to the small number of distributors.

---

> ### Author Response · Authors · 2020-10-02
> **Title**
>
> Which client are you using? Try Transmission so it will support webseeds. I just tried it now and it is working for me.

---

### Decision · Program_Chairs · 2019-12-19

**Decision:**

Reject

**Comment:**

This paper introduces a new ECG dataset. While I appreciate the efforts to clarify several points raised by the reviewers, I still believe this contribution to be of limited interest to the broad ICLR community. As such, I suggest this paper to be submitted to a more specialised venue.